# Classifying Livelihood Strategies Adopting the Activity Choice Approach in Rural China

**Rui Sun \*** , **Jianing Mi, Shu Cao and Xiao Gong**

School of Management, Harbin Institute of Technology, Harbin 150001, China; mijianing@hit.edu.cn (J.M.);
caoshu318@163.com (S.C.); gong_xiao@hotmail.com (X.G.)

**\*** Correspondence: sunrui@stu.hit.edu.cn

**Abstract:** The classification of livelihood strategies is important for designing effective and targeted poverty-reducing strategies. This paper classified livelihood strategies adopting the activity choice approach and compared differences among income levels, asset endowments, poverty rates, and poverty causes of different household clusters to provide bases for the identification of targeted poverty-reducing strategies. By making the two-step cluster analysis, 2042 households were divided into four clusters. Agricultural households get a relatively low income because of the reliance on agricultural production and the lack of required assets to enter more remunerative livelihood strategies. Self-employment is the most remunerative livelihood strategy and high physical and financial capital accumulations are the premise of adopting a self-employed strategy. Featured with a medium-level income and asset endowments, wage-employed households benefit from a more-educated labor force and shoulder a heavy burden caused by children's education at the same time. Besides, rural households face a series of social issues from labor migration, especially self-employed and wage-employed households. Non-labor households have a low-level income and asset endowments with older family members and an unhealthier labor force caused mainly by the aging population and accompanying diseases and disabilities. The transfer income-oriented non-labor households are the main object of poverty alleviation.

**Keywords:** livelihood strategies; livelihood assets; the activity choice approach; pro-poor policies and measures; rural China

---

## 1. Introduction

China has made great progress in poverty alleviation. With the development of the poverty reduction work, units of implementing pro-poor interventions are becoming more and more concentrated from areas to counties, and then to villages, households and individuals. At the same time, criteria for recognizing the poor are getting increasingly comprehensive from single-dimensioned monetary measures to multi-dimensioned monetary and non-monetary ones [1,2]. Multidimensional poverty measurement has been a hot topic since the writing of Sen has laid a conceptual foundation for it [3–5]. Several frameworks were built, among which, the multidimensional poverty index (MPI) occupied an important position as an internationally comparable index reflecting acute poverty by obtaining deprivations in health, education and living standards [2,6,7]. Besides, a series of multidimensional poverty measurement methodologies were developed [7]. For instance, Atkinson contrasted the counting approach and approaches based on social welfare to try to set them in a common framework [8]; Betti et al. proposed the fuzzy approach which regarded poverty as a matter of degree [9]; Alkire and Foster put forward a novel methodology combining the counting approach and the FGT measures [4]; etc.

The sustainable livelihoods framework provides another alternative multidimensional perspective for development and poverty studies. In the framework, people make/undertake a range of choices/activities to achieve livelihood outcomes on the basis of asset endowments. Meanwhile, a battery of internal and external factors will exert important impacts on the system, including vulnerability context and transforming structures and processes [10]. Livelihood strategies denote the range and combination of activities converting possessed livelihood assets into expected livelihood outcomes. The classification of livelihood strategies is essential for revealing different livelihood patterns and designing targeted poverty-reducing interventions.

Approaches of classifying livelihood strategies incorporate the asset-based approach, activity choice approach and income-based approach. The asset-based approach classifies livelihood strategies from the perspective of input according to asset allocation across different activities [11] or asset portfolios [12]. However, it is hard for the asset-based approach to capture nonproductive income-generating activities not involving asset inputs or difficult to measure asset inputs into them, such as investment, retirement, transfer payment, etc. The income-based approach classifies livelihood strategies from the perspective of output according to income from a certain source, for example, nonfarm income [13,14], forest income [15], cash transfer income [16], etc., or income from several sub-divided sources (income composition) [17]. Nevertheless, the income-based approach has its inherent drawbacks. Firstly, the stochastic nature of income could introduce considerable variations into studies; secondly, it fails to present asset and activity differences among households belonging to the same group according to the income-based classification [18,19].

Compared with the asset-based approach and income-based approach, the activity choice approach stems from the definition and essence of livelihood strategies and classifies livelihood strategies from the perspective of the process (see Figure 1). Livelihood strategies connect livelihood assets and livelihood outcomes through a sequence of income-generating activities. Thereby, Nielsen et al. [18] pointed out that activity variables should link the stock concept of assets and the ex-post flow of income, which is ordinarily employed to measure livelihood outcomes [20]. However, there exist two dominant deficiencies in the limited existing studies which adopted the activity choice approach to classify livelihood strategies. Firstly, a "binary" method is usually employed in the process of adopting activity variables, namely, for productive activities, variables measuring the allocation of labor and other inputs were adopted; while for nonproductive activities, which were difficult or impossible to be measured by labor or other inputs, variables representing income were adopted. However, the "binary" division is somewhat incomprehensive and inflexible, which is a "mechanical" combination of the asset-based approach and the income-based approach. Secondly, self-employment was measured by input costs [18,19] or income [21] instead of labor allocation in these studies, for it is difficult and time-consuming to capture the self-employed labor allocation [18], though labor allocation is the most direct measure of how much time households choose to invest in each activity [19].

Besides, studies targeted in broader regions of rural China are limited. There are various livelihood studies conducted in rural China. However, most, if not all, of these studies are targeted in local areas. For example, Fang et al. [14] analyzed the sensitivity of livelihood strategy to livelihood capital in the upper reaches of the Minjiang River; Liao, Barrett and Kassam [22] studied livelihood diversification in Xinjiang; Liu and Liu [23] selected suburban Shanghai as the research object; Wu, Li and Hou [17] and Ding et al. [24] examined determinants and temporal and spatial changes of livelihood patterns in Inner Mongolia; Hua, Yan and Zhang [21] and Yang et al. [25] focused their studies on the Tibetan Plateau; Yang et al. [26] compared livelihood assets and livelihood strategies between two terrace systems located in Yunnan and Hebei Province respectively; Zhang et al. [27] identified multi-level determinants of livelihood strategy choices in Henan Province; etc. Studies targeted in the vast rural areas of China are rare because it is time-, labor- and money-consuming to collect data on a large geographical scale. Fortunately, the implementation of several nation-wide surveys provides valuable data for us and makes studies targeted in broader regions possible.

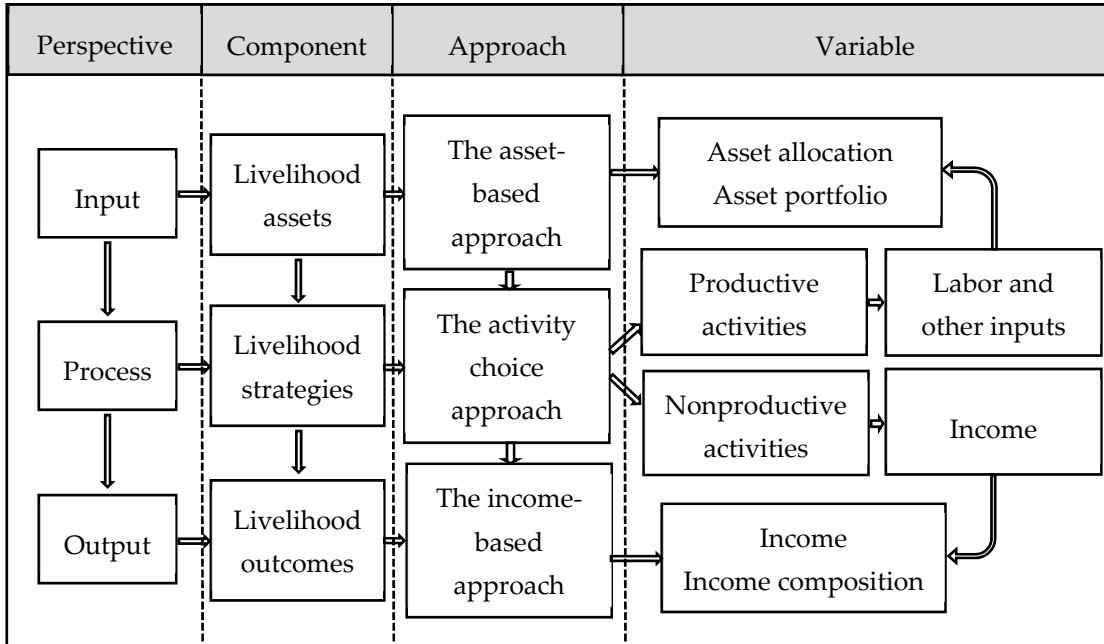

**Figure 1.** Three approaches of classifying livelihood strategies.

This paper supplements the existing studies from the following aspects. Firstly, we use the data from a nation-wide survey to conduct a broader livelihood study targeted in rural China. Secondly, contrary to a "divided" method, we adopt a "combined" method and take asset allocation and income composition into consideration at the same time to generate livelihood strategy clusters from a large set of complementary variables. Thirdly, different from existing studies measuring self-employment with input costs or income, this study measures self-employment with labor allocation directly through interpreting and classifying self-reported occupations of labor force to capture self-employed labor allocation. Fourthly, we interviewed an official to learn more about concrete pro-poor measures implemented at the local level and summarized targeted pro-poor interventions in line with disparate characteristics of different household groups.

The paper aims to answer the following questions: (i) How do rural households make their living? To answer this question, we combined variables representing labor allocation and income composition to classify livelihood strategies and revealed prevailing livelihood patterns in rural China. (ii) What are the differences? This section answers what makes different household groups different. We tested if statistically significant differences existed between every two clusters to reveal income and asset discrepancies among different household groups. (iii) What hinders a household from adopting more remunerative livelihood strategies? We explored determinants of different livelihood strategy options to find barriers of entering a more profitable livelihood strategy; (iv) What can we do? Taking the traits of different household clusters into consideration, we analyzed the most vital causes that different household clusters were stuck in poverty and summarized corresponding pro-poor policies as well as measures combining prevailing policies implemented in rural China and concrete measures conducted at the local level.

To achieve the above goals, we arranged the main content of this paper as below. Section 2 gives a brief introduction to the data and the methods adopted by the study. Section 3 exhibits the results of the two-step cluster analysis, the pairwise comparison, the multinomial logistic regression, and the interview. Section 4 discusses the results further integrating vulnerability contexts and institutional barriers faced by different household clusters. Section 5 gives the conclusions.

## 2. Materials and Methods

### 2.1. Data

This study adopted the family- and individual-level data of 2016 China Labor-force Dynamic Survey (CLDS). We took the household as the basic unit of analysis and the household-level data was the dominant data source of this study. However, we turned to the individual-level data for more detailed information when the description about the occupation of a laborer was ambiguous in the family-level data. In consideration of research purposes and data quality, we conducted a strict screening on the original data. Firstly, rural families were separated from urban families according to whether they lived in village committees (Cunweihui) or neighborhood committees (Juweihui); secondly, families which did not answer certain important items related to the study were deleted; thirdly, families with total income/expenditure being inconsistent with the sum of income/expenditure from sub-component sources and the total area of the grain field being at variance with the sum of the area of the paddy/irrigated grain field and the area of the dry grain field were deleted. Finally, we got 2042 households.

The selected households covered 27 provinces and cities except for Shanghai and Qinghai, which were excluded for the absence of rural samples. The selected households had an average family size of 4.2 and a mean age of household heads of 51.7. In the selected households, male-headed households were 86.2% while only 13.8% of them were female-headed. Sixty-three percent of the selected households had access to tap or pure water and 99.6% of them had access to electricity. The homeownership rate of the chosen households was 88.0%. During 2015, the average income of the chosen households was 40,216.0 RMB, which was close to that (40,603.9 RMB) of the whole sample of rural households surveyed by the 2016 CLDS [28]; the average expenditure of the selected households was 33,683.5 RMB. The average area of the cropland was 7.2 (mu; 1 mu ≈ 666.667 m$^2$; the same below) for the chosen households.

### 2.2. Labor Force, Self-Employment and Activity Variables

#### 2.2.1. Labor Force

In order to calculate labor allocation in different income-generating activities, we needed to define and classify the labor force. This paper defines labor force as household members who engaged in full-time jobs, part-time jobs, temporary jobs and agricultural production during the survey period. As a household was defined as a group of people living under the same roof and sharing resources [19], migrant workers were firstly separated from local workers because they lived apart from the surveyed households and their income was independently controlled except for the part sent back to their families as remittances. For local workers, they were further classified into agricultural workers, wage-employed workers, and self-employed workers according to their descriptions about their occupations. Concretely, (i) if a worker migrated to seek employment, he/she was grouped into migrant workers, and (ii) if he/she did not, he/she would be categorized into one of the other three groups in accordance with his/her detailed description about his/her occupation.

#### 2.2.2. Self-Employment

One of the highlights of this paper is that it measures self-employment with labor allocation, but there is no one universal and broadly-approved definition of self-employment [29]. Though there are distinct divergences among definitions of different organizations and researchers, employers and self-employed individuals without employees are generally included in the self-employed category. In China, self-employment was defined from perspectives of occupational classification [30], capital requirement [31], employment status [32], etc. In this paper, we identified self-employed workers according to occupational classification, incorporating private business owners, shopkeepers, vendors, freelancers (also called independent professionals [29], such as writers, craftsmen, etc.) and

self-employed drivers, builders, decorators, plumbers, electricians, and carpenters. Table 1 lists people of different occupations included in each category of labor force.

**Table 1.** People of different occupations included in each category of labor force.

| Category of Labor Force | People of Different Occupations |
| --- | --- |
| Agricultural workers | plant farmers; livestock breeders; aquaculturists; beekeepers; fishermen; environmental product collectors |
| Wage-employed workers | regular and non-regular employees |
| Self-employed workers | private business owners; shopkeepers; vendors; freelancers; self-employed drivers, builders, decorators, plumbers, electricians and carpenters |
| Migrant workers | workers migrating to seek employment |

### 2.2.3. Activity Variables

A livelihood strategy refers to the combination of income-generating activities [18]. CLDS splits the total income of a household into seven parts, which are, agricultural income, wages, operational income, property income, remittances, pensions, and relief funds. This division covers almost all income-generating activities in rural China, including productive activities, investment activities, transfer payment activities, etc. For productive activities, variables representing labor allocation and income composition were adopted; for nonproductive activities, which involved almost no labor inputs, variables representing income composition were adopted. Since labor migration is tightly related with remittance income, the number of migrant workers was also included in activity variables. Finally, we adopted 11 activity variables: (i) the number of agricultural workers, (ii) the number of wage-employed workers, (iii) the number of self-employed workers, (iv) the number of migrant workers, (v) the percentage of agricultural income in total income, (vi) the percentage of wages in total income, (vii) the percentage of operational income in total income, (viii) the percentage of property income in total income, (ix) the percentage of remittances in total income, (x) the percentage of pensions in total income, and (xi) the percentage of relief funds in total income (see Figure 2).

### 2.3. The Two-Step Cluster Method

This paper used the two-step cluster method to classify livelihood strategies. Compared with the partitional and hierarchical clustering methods, the two-step clustering method is less arbitrary because it adopts the log-likelihood distance as the criterion to define the optimal number of clusters "automatically" and can handle categorical and continuous variables at the same time without pre-continuation of categorical variables [21]. Besides, compared with the basic latent class model normally employed by latent class cluster analyses, it breaks the assumption that there are no covariates and local dependencies between variables [18,19].

### 2.4. The Multinomial Logistic Regression

The multinomial logistic regression is always employed to recognize determinants of livelihood strategies because it is applicable for regression analyses in which dependent variables are categorical ones. This study employed multinomial logistic regression to test the sensitivity of different livelihood strategy choices to livelihood assets because the dependent variable of this study, different livelihood strategy options, is a categorical variable.

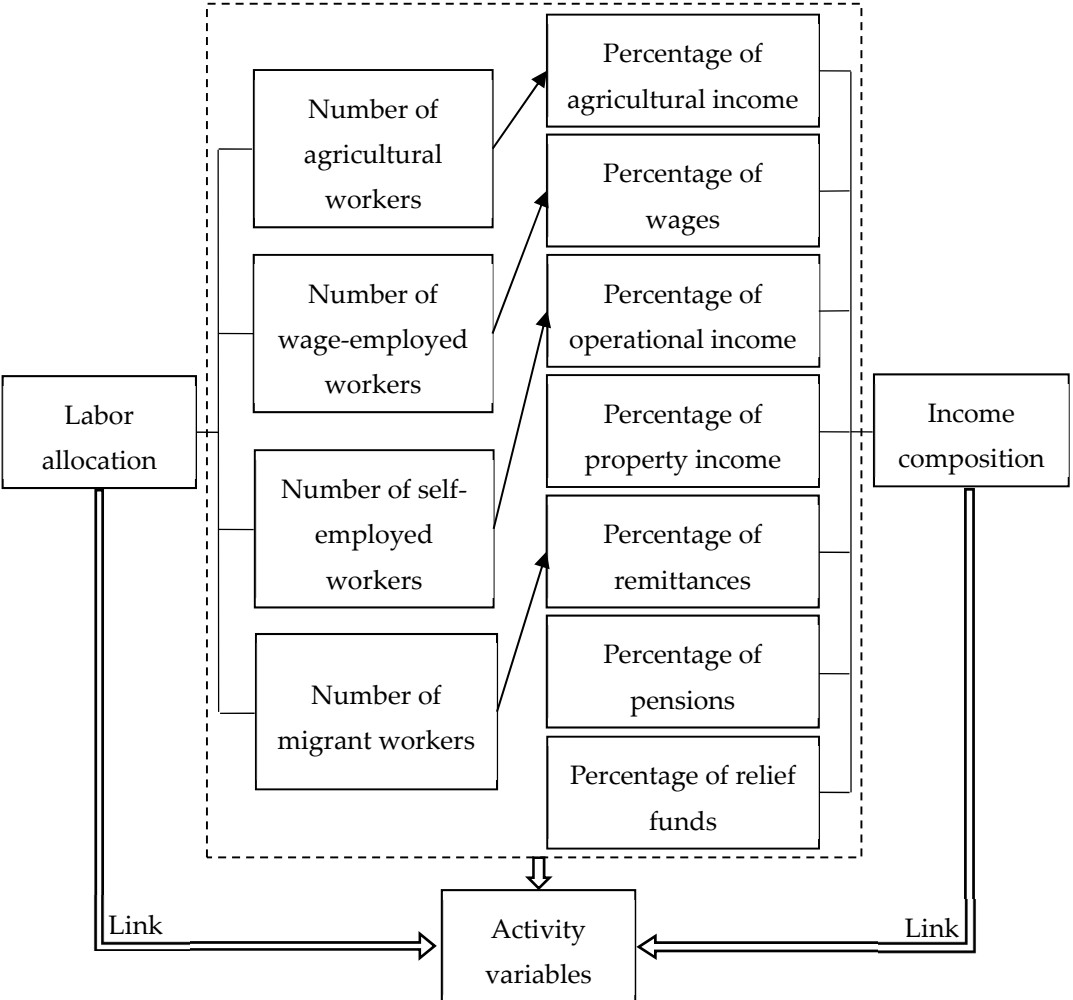

**Figure 2.** Activity variables.

## 2.5. Unstructured Interview

In order to learn more about the concrete pro-poor measures taken at the primary level, we interviewed an official who was the deputy magistrate responsible for the poverty reduction work in a subordinate county of Harbin City, Heilongjiang Province through WeChat. It was an unstructured interview because we did not make a question list to restrict what the official would say. Instead, we merely asked the official to introduce the pro-poor measures taken in her county and then organized what she said.

## 2.6. The Livelihood Capital Index System

The choice of a household's livelihood strategy is dictated by its asset endowment [20]. The sustainable livelihoods framework divides livelihood assets into five types [10]. For natural capital, we firstly divided the cropland into the grain field and non-grain field to represent the allocation of the cropland to grain and non-grain plants [14,17], and then we split the grain field into the paddy/irrigated grain field and the dry grain field further to represent the quality of the grain field [17,23,33,34]. As for human capital, three indicators were adopted, that was, the age of the household head [23,35,36], the education level of the labor force [14,37], and the health condition of the labor force. With respect to physical capital, home ownership, durable goods [38], livestock [12,17] and agricultural implements [15,35] were adopted. For financial capital, we employed income [23,39] and debt [19]. Limited by data availability, only one indicator was adopted to represent social capital,

that was, social spending [17,21], which was a proxy of the cost to sustain a household's relative network (see Table 2).

**Table 2.** Index system of livelihood capital.

| Livelihood Capital | Index [1] | Value Assignment |
|---|---|---|
| Natural capital | Paddy/irrigated grain field (N1) | The area of the paddy/irrigated grain field owned by a household |
| | Dry grain field (N2) | The area of the dry grain field owned by a household |
| | Non-grain field (N3) | The area of the non-grain field, including forest field, orchard, grass field, pond and vegetable field |
| Human capital | Age of household head (H1) | The age of the household head |
| | Education level of labor force (H2) | Illiteracy = 1; Primary school = 2; Middle school = 3; High school and technical secondary school = 4; Junior college and above = 5 Then calculate the average |
| | Health condition of labor force (H3) | Very unhealthy = 1; Unhealthy = 2; General = 3; Healthy = 4; Very healthy = 5 Then calculate the average |
| Physical capital | Home ownership (P1) | Own = 1; Borrow [2] = 2; Rent = 3 |
| | Durable goods [3] (P2) | Possess = 1; Otherwise = 0 Then calculate the average |
| | Agricultural implements (P3) | Possess = 1; Otherwise = 0 |
| | Livestock (P4) | Possess = 1; Otherwise = 0 |
| Financial capital | Income (F1) | The gross income of a household during the year |
| | Debt (F2) | A household owes money = 1; Otherwise = 0 |
| Social capital | Social spending (S1) | The money spent on important social events during the year, such as marriage of relatives |

[1] Appellations of indices are listed in brackets and they will be used to represent capital indices in tables hereafter. [2] "Borrow" refers to living in a house provided by relatives, friends, employment units, governments, etc., freely or at low cost. [3] Durable goods include color TV, air conditioner, refrigerator, washing machine, piano, VCD/DVD, video recorder/camera and desktop/laptop computer.

## 3. Results

### 3.1. Household Clusters

By performing the two-step cluster analysis, 2042 households were divided into four clusters depending on their livelihood strategy options (see Table 3).

The first cluster included 725 households and accounted for 35.50% of the whole sample. As can be seen in Table 3, Cluster 1 had a greater mean value in agricultural workers and a corresponding greater mean value in the percentage of agricultural income. Hence, Cluster 1 was named as "agricultural households". The mean value of agricultural workers for Cluster 1 was 1.606, which was 2.76 to 5.99 times that of the other three clusters. Besides, the percentage of agricultural income of total income was 96.51% for Cluster 1, but the percentages of the other three clusters were 5.34%, 4.18% and 6.01% respectively.

The second cluster covered 224 households and occupied 10.97% of the whole sample. Cluster 2 was named as "self-employed households" because of its greater mean values in self-employed workers and the percentage of operational income in total income. For Cluster 2, the mean value of self-employed workers was 1.174, which was much greater than 0.010 of Cluster 1, 0.020 of Cluster 3 and 0.013 of Cluster 4. Moreover, the percentage of operational income of total income was 68.04% for Cluster 2, while the percentages of the other three clusters were 0.60%, 0.23% and 0.40% respectively.

The third cluster contained 201 households and represented 9.84% of the whole sample. The third cluster was named as "non-labor households" because households subordinating to Cluster 3 made livings not dominantly by productive activities but mainly by nonproductive activities. The third cluster had greater mean values in the percentage of property income, remittances, pensions, and relief funds in total income. For Cluster 3, the proportion of property income was 6.13%, which was far

greater than that of other clusters. The percentage of remittances for Cluster 3 was 36.77%, while that of the other three clusters was between 0.22% and 0.39%. Pensions accounted for 25.21% of the total income of Cluster 3, however, the percentages of the other three clusters were all below 1%. The percentage of relief funds occupied 23.17% for Cluster 3, which was much greater in comparison with 0.21% of Cluster 1, 0.02% of Cluster 2 and 0.07% of Cluster 4.

**Table 3.** Cluster centroids of the two-step cluster analysis.

| Activity Variable | | 1 | 2 | 3 | 4 | Total |
|---|---|---|---|---|---|---|
| Agricultural workers | Mean | 1.606 | 0.268 | 0.303 | 0.581 | 0.883 |
| | Std. Dev. | 0.861 | 0.628 | 0.665 | 0.827 | 0.974 |
| Wage-employed workers | Mean | 0.069 | 0.321 | 0.109 | 0.839 | 0.437 |
| | Std. Dev. | 0.279 | 0.602 | 0.422 | 0.938 | 0.775 |
| Self-employed workers | Mean | 0.010 | 1.174 | 0.020 | 0.013 | 0.140 |
| | Std. Dev. | 0.098 | 0.821 | 0.140 | 0.115 | 0.465 |
| Migrant workers | Mean | 0.728 | 0.482 | 0.756 | 1.001 | 0.823 |
| | Std. Dev. | 0.989 | 0.825 | 1.306 | 1.130 | 1.084 |
| Agricultural income | Mean | 0.965 | 0.053 | 0.042 | 0.060 | 0.379 |
| | Std. Dev. | 0.114 | 0.160 | 0.121 | 0.123 | 0.452 |
| Wages | Mean | 0.023 | 0.258 | 0.043 | 0.929 | 0.446 |
| | Std. Dev. | 0.095 | 0.398 | 0.142 | 0.131 | 0.464 |
| Operational income | Mean | 0.006 | 0.680 | 0.002 | 0.004 | 0.079 |
| | Std. Dev. | 0.052 | 0.404 | 0.025 | 0.036 | 0.253 |
| Property income | Mean | 0.000 | 0.001 | 0.061 | 0.002 | 0.007 |
| | Std. Dev. | 0.002 | 0.014 | 0.214 | 0.015 | 0.070 |
| Remittances | Mean | 0.004 | 0.003 | 0.368 | 0.002 | 0.039 |
| | Std. Dev. | 0.034 | 0.032 | 0.450 | 0.024 | 0.180 |
| Pensions | Mean | 0.000 | 0.003 | 0.252 | 0.002 | 0.026 |
| | Std. Dev. | 0.004 | 0.029 | 0.417 | 0.023 | 0.151 |
| Relief funds | Mean | 0.002 | 0.000 | 0.232 | 0.001 | 0.024 |
| | Std. Dev. | 0.023 | 0.002 | 0.400 | 0.009 | 0.144 |

The fourth cluster was named "wage-employed households" and contained 892 households. It occupied 43.68% of the whole sample and had greater mean values in wage-employed workers, migrant workers and the percentage of wages. For Cluster 4, the mean value of wage-employed workers was 0.839, while that of the other three clusters was 0.069, 0.321 and 0.109 respectively. As for migrant workers, the mean value of Cluster 4 was 1.001, but that of the other three clusters was 0.728, 0.482 and 0.756 respectively. The percentage of wages of total income was 92.92% for Cluster 4, while the percentages of the other three clusters were 2.26%, 25.82% and 4.32%.

Labor allocation and income composition of each household group correspond well with their respective dominant livelihood activities, which implies a sound result of the two-step cluster analysis.

*3.2. Income and Livelihood Capital of Different Household Clusters*

3.2.1. Income of Different Household Clusters

There existed distinct differences among income and income composition of different household clusters (see Table 4, Figure 3 and Table A1).

According to Table 4, Figure 3 and Table A1, self-employed households got an average total income of 80,741.61 RMB, which was significantly higher than that of other clusters, indicating self-employment was the most remunerative livelihood strategy among these four livelihood strategies. As for income composition, self-employed households derived 66.00% of the total income from operational activities, followed by wage employment and agricultural production with 30.08% and 3.35%. The percentages of income from other sub-component sources were all below 1%. According to Table A1, on average, operational income of self-employed households (53,289.29 RMB) was significantly higher than that of

other clusters, and the operational income of the other three clusters exhibited no significant differences with 227.59 RMB, 79.60 RMB and 284.75 RMB respectively.

**Table 4.** Total and subdivided income of different household clusters.

| Income | 1 | | 2 | | 3 | | 4 | |
|---|---|---|---|---|---|---|---|---|
| | Mean | Std. Dev. | Mean | Std. Dev. | Mean | Std. Dev. | Mean | Std. Dev. |
| Total income | 22,317.77 | 37,351.637 | 80,741.61 | 108,942.565 | 20,348.91 | 28,760.766 | 49,063.27 | 46,490.748 |
| Agricultural income | 21,096.06 | 36,287.396 | 2700.89 | 10,229.895 | 575.12 | 1875.254 | 2636.09 | 9369.424 |
| Wages | 886.34 | 5244.918 | 24,288.39 | 77,661.913 | 2937.11 | 15,535.070 | 45,684.27 | 44,759.327 |
| Operational income | 227.59 | 1944.481 | 53,289.29 | 83,783.278 | 79.60 | 820.751 | 284.75 | 3421.342 |
| Property income | 13.79 | 371.391 | 92.86 | 785.544 | 1629.90 | 7609.013 | 145.85 | 1482.882 |
| Remittances | 78.62 | 970.930 | 108.93 | 1048.044 | 4077.11 | 9234.255 | 68.95 | 775.617 |
| Pensions | 1.10 | 29.711 | 250.00 | 2850.537 | 9801.89 | 20,316.978 | 213.70 | 2371.116 |
| Relief funds | 14.26 | 159.527 | 11.25 | 134.232 | 1248.16 | 3831.728 | 29.66 | 417.487 |

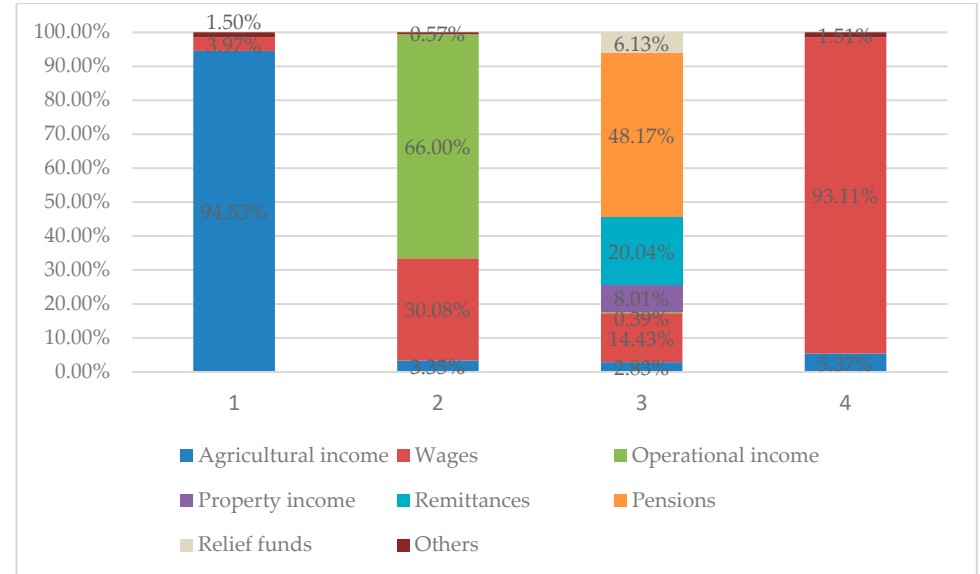

**Figure 3.** Income composition of different household clusters. Some percentages were added up and presented with "Others" for Cluster 1, Cluster 2 and Cluster 4 in consideration of a clear layout of Figure 3. For Cluster 1, "Others" included the percentage of operational income (1.02%), property income (0.06%), remittances (0.35%), pensions (0.00%) and relief funds (0.06%). For Cluster 2, "Others" included the percentage of property income (0.12%), remittances (0.13%), pensions (0.31%) and relief funds (0.01%). For Cluster 4, "Others" included the percentage of operational income (0.58%), property income (0.30%), remittances (0.14%), pensions (0.44%), and relief funds (0.06%).

Wage-employed households got the second highest total income with an average total income of 49,063.27 RMB, which was apparently higher than that of agricultural households and non-labor households. Wages accounted for 93.11% of wage-employed households' total income, followed by agricultural income with 5.37%. The percentages of income generated from operational and nonproductive activities were below 1%. As can be seen in Table A1, the wages of wage-employed households (45,684.27 RMB) was significantly higher than that of self-employed households (24,288.39 RMB), which was apparently higher than that of non-labor households (2937.11 RMB) and agricultural households (886.34 RMB) successively. However, there was no significant difference between non-labor and agricultural households with respect to wages, on average.

Agricultural households got a relatively low total income with an average of 22,317.77 RMB. It was higher than that of non-labor households but the difference was not statistically significant. Agricultural income occupied 94.53% of the total income, followed by wages and operational income with 3.97% and 1.02% respectively. The percentages of income from other sub-component sources

were below 1%. According to the result of the pairwise comparison, agricultural income of agricultural households (21,096.06 RMB) was significantly higher than that of other clusters. At the same time, the value of non-labor households (575.12 RMB) was obviously lower than that of self-employed and wage-employed households, which was 2700.89 RMB and 2636.09 RMB respectively.

Non-labor households got the lowest total income with a mean value of 20,348.91 RMB. For non-labor households, pensions constituted 48.17% of the total income, followed by remittances, wages, property income, relief funds, agricultural income, and operational income with 20.04%, 14.43%, 8.01%, 6.13%, 2.83%, and 0.39%, respectively. For property income, remittances, pensions, and relief funds, the same conclusion for the pairwise comparison was reached, that the value of non-labor households was significantly higher than that of the other three clusters and there existed no distinct differences among the other three clusters on average, with only one exception that pensions of agricultural households were significantly lower than that of wage-employed households.

### 3.2.2. Livelihood Capital of Different Household Clusters

Table 5 shows the mean values and standard deviations of livelihood capital of different household clusters, and Table A2 shows the result of the pairwise comparison.

**Table 5.** Livelihood capital of different clusters.

| Livelihood Capital | 1 | | 2 | | 3 | | 4 | |
|---|---|---|---|---|---|---|---|---|
| | Mean | Std. Dev. | Mean | Std. Dev. | Mean | Std. Dev. | Mean | Std. Dev. |
| N1 | 2.43 | 4.812 | 1.21 | 2.814 | 0.99 | 1.751 | 1.47 | 2.823 |
| N2 | 6.40 | 12.499 | 1.44 | 4.094 | 1.03 | 2.385 | 1.75 | 4.095 |
| N3 | 3.16 | 18.130 | 2.74 | 14.475 | 1.43 | 7.361 | 1.30 | 7.091 |
| H1 | 52.93 | 11.509 | 46.72 | 11.048 | 65.14 | 12.913 | 49.01 | 10.658 |
| H2 | 2.39 | 0.949 | 2.93 | 0.949 | 1.44 | 1.552 | 2.84 | 0.958 |
| H3 | 3.41 | 1.192 | 3.89 | 1.032 | 1.93 | 1.959 | 3.76 | 1.053 |
| P1 | 1.05 | 0.224 | 1.37 | 0.727 | 1.19 | 0.441 | 1.20 | 0.543 |
| P2 | 0.37 | 0.170 | 0.55 | 0.175 | 0.36 | 0.213 | 0.46 | 0.193 |
| P3 | 0.28 | 0.449 | 0.05 | 0.217 | 0.04 | 0.196 | 0.09 | 0.279 |
| P4 | 0.16 | 0.371 | 0.03 | 0.174 | 0.06 | 0.247 | 0.03 | 0.180 |
| F1 | 22,317.77 | 37,351.638 | 80,741.61 | 108,942.565 | 20,348.91 | 28,760.766 | 49,063.27 | 46,490.748 |
| F2 | 0.39 | 0.488 | 0.29 | 0.455 | 0.22 | 0.418 | 0.32 | 0.467 |
| S1 | 2895.01 | 4724.491 | 4311.83 | 8478.684 | 1308.71 | 2937.377 | 2760.99 | 4325.080 |

According to Table 5, the area of the paddy/irrigated grain field for agricultural households was 2.43, followed by the other three clusters with 1.21, 0.99 and 1.47 respectively. As for the dry grain field, areas of different clusters were 6.40, 1.44, 1.03 and 1.75 respectively. As can be seen in Table A2, for both the paddy/irrigated and dry grain field, the area of agricultural households was significantly greater than that of the other three clusters, and the area of non-labor households was much smaller than that of wage-employed households. In terms of the non-grain field, the area of agricultural households was 3.16, followed by the other three clusters with 2.74, 1.43 and 1.30 respectively. There were no significant differences among the four clusters, on average.

For the matter of human capital, the age of the household head for self-employed households was 46.72 while that of wage-employed, agricultural and non-labor households was 49.01, 52.93 and 65.14 successively. On average, differences between every two clusters had statistical significance. Education levels of the labor force for self-employed and wage-employed households were 2.93 and 2.84, which were significantly higher than those of agricultural and non-labor households. Meanwhile, education levels of the labor force for agricultural and non-labor households were 2.39 and 1.44, and the former was significantly higher than the latter on average. Similarly, health conditions of the labor force for self-employed and wage-employed households were 3.89 and 3.76, which were significantly better than those of agricultural and non-labor households, and health conditions of the labor force for agricultural and non-labor households were 3.41 and 1.93 with the former being significantly better than the latter.

For physical capital, the home ownership of agricultural households was 1.05, followed by non-labor, wage-employed and self-employed households with 1.19, 1.20 and 1.37 respectively. Differences between every two clusters were statistically significant. With respect to durable goods, ownership rates of self-employed, wage-employed, agricultural, and non-labor households were 0.55, 0.46, 0.37, and 0.36 respectively. On average, self-employed households had a noticeably higher ownership rate compared with other clusters, and the ownership rate of wage-employed households was significantly higher than that of agricultural and non-labor households. The ownership rate of agricultural implements for agricultural households was 0.28, which was significantly higher than that of other clusters. Similarly, the ownership rate of livestock for agricultural households was 0.16 and it was apparently higher than that of the other clusters.

For financial capital, the ratio of households having debt was 0.39 for agricultural households, followed by wage-employed, self-employed, and non-labor households with 0.32, 0.29 and 0.22, respectively. The ratio of agricultural households was significantly higher than that of other clusters. Besides, the ratio of wage-employed households was apparently higher than that of non-labor households.

For social capital, the spending on cash gifts was 4311.83 RMB for self-employed households, while that of the other three clusters was 2895.01 RMB, 2760.99 RMB and 1308.71 RMB successively. On average, non-labor households spent less money on cash gifts than other clusters, and at the same time, self-employed households spent more money in comparison with wage-employed households.

### 3.3. Determinants of Different Livelihood Strategy Options

This paper identified determinants of different livelihood strategy options employing the multinomial logistic regression, and Table 6 shows the result of the analysis.

**Table 6.** Determinants of different livelihood strategy options.

| Livelihood Capital | | Household Cluster | | | | | |
|---|---|---|---|---|---|---|---|
| | | Self-Employed Households | | Non-Labor Households | | Wage-Employed Households | |
| | | COEF | EXP(B) | COEF | EXP(B) | COEF | EXP(B) |
| N | N1 | −0.606 *** | 0.545 | −0.651 *** | 0.521 | −0.475 *** | 0.622 |
| | N2 | −0.898 *** | 0.407 | −1.662 *** | 0.190 | −0.825 *** | 0.438 |
| | N3 | 0.047 | 1.048 | −0.055 | 0.947 | −0.206 * | 0.814 |
| H | H1 | −0.305 ** | 0.737 | 0.835 *** | 2.305 | −0.250 *** | 0.779 |
| | H2 | 0.071 | 1.074 | 0.019 | 1.019 | 0.299 *** | 1.349 |
| | H3 | 0.072 | 1.075 | −0.465 *** | 0.628 | −0.040 | 0.961 |
| P | P1 = 1 | −4.152 *** | 0.016 | −2.717 * | 0.066 | −3.070 ** | 0.046 |
| | P1 = 2 | −3.474 ** | 0.031 | −1.885 | 0.152 | −2.715 * | 0.066 |
| | P2 | 0.651 *** | 1.918 | 0.339 ** | 1.404 | 0.149 * | 1.161 |
| | P3 = 0 | 1.346 *** | 3.843 | 0.870 * | 2.387 | 0.914 *** | 2.494 |
| | P4 = 0 | 0.684 | 1.981 | 0.311 | 1.365 | 1.124 *** | 3.077 |
| F | F1 | 1.796 *** | 6.027 | 0.751 ** | 2.118 | 1.636 *** | 5.134 |
| | F2 = 0 | 0.219 | 1.245 | 0.066 | 1.068 | 0.172 | 1.187 |
| S | S1 | 0.094 | 1.099 | −0.309 | 0.734 | −0.073 | 0.930 |
| Constant | | 0.866 | - | −0.366 | - | 1.586 | - |
| LR chi$^2$ = 1238.772 *** (df = 42) | | | | | | | |
| Nagelkerke R$^2$ = 0.500 | | | | | | | |

Note: "***", "**" and "*" represent significant levels at 0.1%, 1% and 5%, respectively.

As can be observed in Table 6, the shrinkage of the cropland was a key driving factor for households to turn to off-farm strategies [21]. The decrease of the paddy/irrigated grain field by one unit increased the probabilities of adopting self-employed, non-labor and wage-employed strategies by 0.545, 0.521 and 0.622 times respectively. Similarly, the reduction of the dry grain field by one unit increased the likelihood of adopting the other three strategies by 0.407, 0.190 and 0.438 times respectively. Compared with the dry grain field, the paddy/irrigated grain field exerted a greater effect (with greater odds ratio values), indicating that the quality of the grain field, as well as the quantity of the grain field, affected the transformation from on-farm to off-farm strategies.

Human capital played an important role in the strategy option. If the age of the household head increased by one unit, the ratios of adopting self-employed and wage-employed strategies decreased by 0.737 and 0.779 times respectively, indicating that a younger household head was more inclined to lead a household to seek off-farm employment [23,27], especially wage employment, which ordinarily imposed strict restrictions on the age of the employees. However, the increase of the age of the household head by one unit made the feasibility of adopting a non-labor strategy increase by 2.305 times. Besides, if the education level of the labor force increased by one unit, the probability of adopting a wage-employed strategy increased by 1.349 times, indicating that a low-educated labor force was a barrier for adopting a wage-employed strategy. The decrease of the health condition of the labor force by one unit would increase the probability of adopting a non-labor strategy by 0.628 times, illustrating the lack of a healthy labor force, caused mainly by the aging population and its accompanying diseases and disabilities, was a vital determinant of adopting a non-labor strategy.

Physical capital exhibited significant differences among different clusters. It can be seen in Table 6 that agricultural households had a higher level of home ownership compared with other clusters. That was probably because agricultural households lived by farmland and were more likely to live in their registered residences where farmland and homesteads were distributed and it was low-cost to build/buy a house. From Table 7, we can see that the percentage of household heads whose registered residences were in accordance with their current residences was 97.51% for agricultural households, while that of the other three clusters was 88.79%, 92.04% and 90.24% respectively. Besides, agricultural households had a significantly lower ownership rate for durable goods compared with other clusters, which coincided with the conclusion that households with higher physical capital holdings were more apt to engage in more remunerative strategies [19]. At the same time, agricultural households had an eminently higher ownership rate for agricultural implements in comparison with other clusters.

**Table 7.** Registered residences of household heads for different household clusters.

| Registered Residence | 1 | 2 | 3 | 4 | Total |
|---|---|---|---|---|---|
| Villages living in | 706 (97.51%) | 198 (88.79%) | 185 (92.04%) | 804 (90.24%) | 1893 (92.84%) |
| Other villages | 8 (1.10%) | 1 (0.45%) | 7 (3.48%) | 10 (1.12%) | 26 (1.28%) |
| Other counties | 6 (0.83%) | 3 (1.35%) | 6 (2.99%) | 3 (0.34%) | 18 (0.88%) |
| Other towns | 4 (0.55%) | 21 (9.42%) | 3 (1.49%) | 74 (8.31%) | 102 (5.00%) |
| Total | 724 | 223 | 201 | 891 | 2039 |

Note: Results reported absolute values and relative values (in brackets).

With respect to financial capital, the added income of one unit would increase the probabilities of employing self-employed, wage-employed and non-labor strategies by 6.027, 5.134 and 2.118 times respectively. This is in line with the general agreement that the financial capital (non-farm earnings, deposit, regular remittance, etc.) possessed by rural households has been a catalyst in increasing the opportunity of off-farm activities [14,21].

### 3.4. Poverty Causes and Targeted Pro-Poor Policies and Measures

The classification of livelihood strategies is conducive to the development and implementation of targeted poverty alleviation. Different household clusters have distinct income levels and asset endowments, which results in disparate poverty rates and poverty causes amongst household clusters. Table 8 shows the most important causes of getting stuck in poverty for poverty-stricken households of different household clusters.

**Table 8.** The most important causes of getting stuck in poverty.

| The Most Important Cause | 1 | 2 | 3 | 4 | Total |
|---|---|---|---|---|---|
| The agricultural income is low and there are no other sources of income | 25 (45.45%) | 5 (38.46%) | 13 (27.66%) | 26 (34.21%) | 69 (36.13%) |
| Sick or disabled family members | 17 (30.91%) | 4 (30.77%) | 16 (34.04%) | 18 (23.68%) | 55 (28.80%) |
| The burden of children's education is heavy | 4 (7.27%) | 1 (7.69%) | 1 (2.13%) | 13 (17.11%) | 19 (9.95%) |
| Poor natural conditions | 0 (0.00%) | 0 (0.00%) | 1 (2.13%) | 1 (1.32%) | 2 (1.05%) |
| The burden to support the old is heavy | 1 (1.82%) | 0 (0.00%) | 2 (4.26%) | 0 (0.00%) | 3 (1.57%) |
| The burden to raise children is heavy | 0 (0.00%) | 0 (0.00%) | 0 (0.00%) | 6 (7.89%) | 6 (3.14%) |
| The lack of labor force | 4 (7.27%) | 1 (7.69%) | 13 (27.66%) | 7 (9.21%) | 25 (13.09%) |
| Natural disasters and emergencies | 3 (5.45%) | 0 (0.00%) | 0 (0.00%) | 0 (0.00%) | 3 (1.57%) |
| Poor traffic conditions | 0 (0.00%) | 0 (0.00%) | 0 (0.00%) | 1 (1.32%) | 1 (0.52%) |
| The lack of enrichment information | 0 (0.00%) | 1 (7.69%) | 0 (0.00%) | 0 (0.00%) | 1 (0.52%) |
| Others | 1 (1.82%) | 1 (7.69%) | 1 (2.13%) | 4 (5.26%) | 7 (3.66%) |
| Total | 55 | 13 | 47 | 76 | 191 |

Note: Results reported absolute values and relative values (in brackets).

Combining income levels, asset endowments and poverty causes of different household clusters, we summarized efficacious pro-poor policies and measures implemented in rural China. Furthermore, in order to learn more about the specific pro-poor measures implemented at the local level, we interviewed a poverty alleviation official from a county of Heilongjiang Province through WeChat for more specific information. Table 9 shows the pro-poor policies and measures targeted in different household clusters.

For non-labor households, it should be noted that they could have been subdivided into two groups though they were categorized into the same cluster by the cluster analysis and there were some noticeable similarities among them. The first subgroup is called "capital-oriented non-labor households" because it contains 71 households which generated income dominantly by capital inputs, including pensions from previous human capital inputs and property income from physical or financial capital inputs. The second subgroup was named "transfer income-oriented non-labor households" because it contains 130 households which lived mainly by transfer income, covering remittances from relatives/friends and relief funds from governments/organizations. The capital-oriented non-labor households had an average income of 40, 475.35 RMB, while that of the transfer income-oriented non-labor households was 9356.77 RMB. The large income inequality led to distinct poverty status between these two subgroups, and the subdivision was beneficial for increasing the target accuracy of pro-poor policies and measures.

According to Table 9, 7.60% of agricultural households were confirmed as poverty-stricken households. When asked "What do you think is the most important cause leading to poverty of your family", 45.45% of agricultural poverty-stricken households chose "the agricultural income is low and there are no other sources of income". Thereby, it was essential to increase income from agricultural production or accelerate transformation from on-farm to off-farm strategies. Some effective policies and measures were implemented. Firstly, with the Grain for Green Policy implemented in rural China, many households leased their land to local governments and they received cash compensation equal to, if not more than, their previous income during the leasing period. Through this,

households were guaranteed a steady income from their cropland, and the labor force which engaged in agricultural production previously could turn to pursue off-farm jobs. Secondly, the prevailing courtyard economy increased agricultural income greatly. As the official told us, some households planted crops/vegetables/fruits labeled as green and healthy in or around their courtyards and some products had developed into branded agricultural products. Thirdly, the village committee provided guidance and assistance for households to adjust the planting structure, which improved the yield through shifting cultivation on one hand and was conducive to arranging planting activities according to the need of the market on the other hand. For example, as the official said, planting potatoes was encouraged in her village, for there was a potato processing factory in the village and the village committee had reached a deal with a factory in Shanghai to provide potatoes for it.

Apart from that, to promote employment and entrepreneurship in rural areas, some actions were taken. As the official said, the village committee would help those who failed to find jobs in local areas seek employment opportunities outside. At the same time, to encourage entrepreneurship in rural areas, banks and rural credit cooperatives provided petty loans at low or zero interest rates to encourage households to start their own businesses.

Of the agricultural poverty-stricken households, 30.91% chose "sick or disabled family members" as the most important reason of getting stuck in poverty. How to improve the rural healthcare service system and provide affordable healthcare in rural areas was a considerable question. The implementation of the new rural cooperative medical service relieved the healthcare burden to a large extent. By 2013, the coverage of the new rural cooperative medical service had reached 98.9% [40], and the percentage of personal healthcare expenditures in total healthcare expenditures had reduced to 33.9% [41]. As the official told us, the local government would pay the new rural cooperative medical insurance for poverty-stricken households and they could apply for a reimbursement of between 75% and 95% of their expenses within the reimbursement scope. Besides, some hospitals would give free medical consultation and offer free medical treatment for villagers.

For self-employed households, the percentage of poverty-stricken households (5.80%) was the lowest, and 38.46% of poverty-stricken households chose "the agricultural income is low and there are no other sources of income" as the most important reason for being trapped in poverty, followed by "sick or disabled family members" with 30.77%.

The percentage of poverty-stricken households of capital-oriented non-labor households was 7.04%, while that of transfer income-oriented non-labor households was 32.31%. As the aging population problem was universal for non-labor households, there existed no wide differences between their choices of the most important poverty causes. Besides "sick or disabled family members" and "the agricultural income is low and there are no other sources of income", "the lack of labor force" was another major cause of poverty for non-labor households. Thereby, it was important to take measures to help households with limited or no labor force out of poverty. Fortunately, a series of actions were taken. As the official mentioned, for households with a limited labor force, the village committee would provide public service jobs for their members who were able to engage in simple work. By this, they did not merely increase income but also had more time to take care of their disabled or sick family members. However, for households with no labor force, the sole option was to provide money and materials for them. One key source of income was dividends from village collective projects. As the official said, the village collective project had already covered every household in her village and paid dividends for three times (1500 RMB/household for the first time, 240 RMB/person for the second time and 200 RMB/person for the third time). Besides, households with no labor force could rent their land out to large-scale grain-production households to get rent. Thirdly, the central and local government would provide allowances for households whose income was below the rural residents' minimum living security standard.

**Table 9.** Pro-poor policies and measures targeted in different household clusters.

| Household Cluster | | Poverty Rate | The Most Important Reason of Getting Stuck in Poverty (>10%) | Enlightenment | Policy and Measure |
|---|---|---|---|---|---|
| Agricultural households | | 7.60% | (i) the agricultural income is low and there are no other sources of income. (ii) sick or disabled family members. | (i) increase income from agricultural production. (ii) accelerate transformation from on-farm to off-farm strategies. (iii) provide affordable healthcare for rural households | (i) the Grain for Green Policy. (ii) courtyard economy. (iii) adjust planting structure (iv) employment assistance (v) rural microfinance. (vi) New Rural Cooperative Medical System (vii) free medical consultation and treatment |
| Self-employed households | | 5.80% | | | |
| Non-labor households | Capital-oriented non-labor households | 7.04% | (i) sick or disabled family members. (ii) the agricultural income is low and there are no other sources of income. (iii) the lack of labor force. | provide living support for households having limited or no labor force | (i) provide public service jobs for households having limited labor force (ii) village collective projects (iii) agricultural land transfer (iv) subsistence security system |
| | Transfer income-oriented non-labor households | 32.31% | | | |
| Wage-employed households | | 8.52% | (i) the agricultural income is low and there are no other sources of income. (ii) sick or disabled family members. (iii) the burden of children's education is heavy | relieve the burden caused by children's education | (i) provide partial or total tuition & fee waivers for students from poverty-stricken households (ii) provide living subsidies for students from poverty-stricken households (iii) poor students' subsidies (iv) student loans |

It was a little surprising that wage-employed households had a higher proportion of low-income families compared to agricultural households. Compared with other clusters, "the burden of children's education is heavy" was another major determinant leading to poverty of wage-employed households. Firstly, that was probably because compared with agricultural and non-labor households, wage-households were relatively "young" and they needed to bear the burden of children's education; secondly, compared with self-employed households, wage-employed households got a relatively low income which made the burden of children's education heavy for them; thirdly, as direct beneficiaries of education, wage-employed households tended to pay more attention to children's education. It was important to relieve the burden caused by children's education for poverty-stricken households. As the official told us, except for a total tuition & fee waiver, the local government would provide living subsidies for students from poverty-stricken households. A middle school/high school/technical secondary school student would receive 2000 RMB per year and the standard for a kindergarten/primary school student was 1000 RMB per year. For college/university students, they could apply for student loans and poor students' subsidies in college/university.

## 4. Discussion

Though protected by a string of pro-poor policies and measures, rural households still face some unavoidable risks and barriers. Rural households diversify their income-generating activities to smooth income, accumulate wealth, and reduce risk exposure [18]. In Table 10, we calculated the number and proportion of households engaging in non-dominant productive activities as a simple proxy of livelihood diversity.

**Table 10.** Number and proportion of households engaging in non-dominant productive activities.

| Livelihood Cluster | Productive Activities | | | |
| --- | --- | --- | --- | --- |
| | Agricultural Production | Wage-Employment | Self-Employment | Migrant Employment |
| 1 | - | 45 (6.21%) | 7 (0.97%) | 325 (44.83%) |
| 2 | 43 (19.20%) | 57 (25.45%) | - | 71 (31.70%) |
| 3 | 42 (20.90%) | 17 (8.64%) | 4 (1.99%) | 72 (35.32%) |
| 4 | 354 (39.69%) | - | 12 (1.35%) | 506 (56.73%) |

Note: Results reported absolute values and relative values (in brackets).

As can be seen in Table 10, except for migrant employment, 6.21% of agricultural households engaged in wage-employment and only 0.97% of them engaged in self-employment, indicating agricultural households had a simple livelihood [17]. The high reliance on agricultural production made agricultural households more prone to natural disasters and emergencies. From Table 8, we can see that 5.45% of poverty-stricken agricultural households chose "natural disasters and emergencies" as the most important cause of poverty. What is more, the low quality of the cropland made agricultural households more sensitive to natural disasters. As can be seen in Table 5, the area of the dry grain field was 2.63 times that of the paddy/irrigated grain field for agricultural households, which made them more sensitive to natural disasters, like drought.

Besides, the conventional agricultural production mode made agricultural households' agricultural production inefficient. Since the household responsibility system was implemented in 1979 [17], the small-scale operation has become a salient feature of the agricultural production mode in rural China [42]. According to our data, even for agricultural households, the average area of the cropland was only 12.00. With the implementation of the household responsibility system, farmers were entitled to possess, utilize, profit from, and dispose of the land [43]. However, land fragmentation caused by the small-scale operation mode exerted a negative influence on mechanized and modernized agricultural production. According to our data, 1176 households engaged in agricultural production among the 2042 households while only 249 households (21.17%) achieved complete mechanization. The high

reliance on agricultural production and the low efficiency of agricultural production brought about a low level of income of agricultural households.

The aging population problem is exerting a great effect on rural China, especially for non-labor and agricultural households. Non-labor households constituted 9.84% of the 2042 households, and aged family members and unhealthy labor force were key determinants of adopting a non-labor strategy. Based on our data, 70.65% of non-labor households did not participate in other productive activities, except some of their children engaged in migrant employment, and 47.76% of them did not engage in any productive activities. For those which engaged in productive activities, 20.90% of them engaged in simple agricultural production, cultivating a small quantity of croplands to meet their own needs for grains [21]. The extremely low participation rate in productive activities caused mainly by the aging population and accompanying diseases and disabilities made non-labor households extremely prone to external and internal shocks and more likely to get stuck in poverty. In total, 61.70% of poverty-stricken non-labor households regarded "sick or disabled family members" or "the lack of labor force" as the most important cause of poverty. Agricultural households, as a relatively "old" household group, were also threatened by the aging population problem.

With respect to self-employed households, 25.45% of them engaged in wage-employment and 19.20% of them engaged in agricultural production. As for wage-employed households, 39.69% of them engaged in agricultural production and 1.35% of them engaged in self-employment. The relatively high level of livelihood diversity made self-employed and wage-employed households less vulnerable to external and internal shocks, but they still faced certain institutional barriers. Labor migration has become a universal phenomenon in rural China, but the household registration system implemented in China constitutes the greatest institutional obstacle hindering the migration of rural workers [44]. As can be seen in Table 7; Table 11, non-agricultural households were more apt to migrate as households compared with agricultural households, especially self-employed households, while individual migration was more common for wage-employed and agricultural households, especially for wage-employed households. However, the household registration system set a lot of obstacles to social inclusion of migrant workers and caused a series of social problems, such as the housing problem, the healthcare and insurance problem, and the education problem of their children.

**Table 11.** Migrant workers of different household clusters.

| Number of Migrant Workers | 1 | 2 | 3 | 4 | Total |
|---|---|---|---|---|---|
| 0 | 400 (55.17%) | 153 (68.30%) | 130 (64.68%) | 386 (43.27%) | 1069 (52.35%) |
| 1 | 175 (24.14%) | 43 (19.20%) | 26 (12.94%) | 239 (26.79%) | 483 (23.65%) |
| 2 | 116 (16.00%) | 21 (9.38%) | 27 (13.43%) | 186 (20.85%) | 350 (17.14%) |
| ≥3 | 34 (4.69%) | 7 (3.13%) | 18 (8.96%) | 81 (9.08%) | 140 (6.86%) |
| Total | 725 | 224 | 201 | 892 | 2042 |

Note: Results reported absolute values and relative values (in brackets).

Firstly, how to solve the housing problem of migrant workers is a thought-provoking issue. According to the study of Yang, the housing of migrant workers was dominated by rental housing and they had an extremely low homeownership rate and were excluded from the affordable housing system [45]. "Living conditions" ranked second among the 10 least satisfactory public services and "providing affordable housing" ranked third among the nine main appeals of migrant workers to the government according to a survey of Development Research Center of the State Council [46]. Secondly, migrant workers faced more difficulties with respect to children's education. How to provide equal educational opportunities for children migrating with their parents deserves attention. Limited by the household registration system, enrollment in urban public schools is hard for migrant children. As a prerequisite to enrollment, urban public schools commonly require an excessive amount of documentation and charge expensive fees which make public education inaccessible to migrant households [47]. Last but not least, the household registration system excludes migrant workers from the urban health insurance system. Migrant workers are confronted with poor medical conditions

and expensive medical costs in urban areas. Since the new rural cooperative medical system takes the county as the basic unit, remote settlement is difficult. What is more, the percentage of employed units paying insurance for migrant workers is low. According to the survey of Development Research Center of the State Council, "social insurance" and "medical conditions" ranked third and fourth correspondingly among the 10 least satisfactory public services, and at the same time, "improving social insurance" and "improving medical conditions" ranked second and fourth respectively among the nine main appeals of migrant workers to the government [46].

## 5. Conclusions

This paper classified 2042 households into four clusters via adopting the activity choice approach. Agricultural households got a relatively low income because they had a simple livelihood and were trapped in conventional inefficient agricultural production. For agricultural households, it was of importance to increase agricultural income or accelerate transformation from on-farm to off-farm strategies. Self-employment was the most remunerative strategy which required comparatively high physical and financial capital accumulations. However, because of the universal labor migration phenomenon and the household registration system, a series of social problems appeared, such as the housing problem, the healthcare and insurance problem, and the education problem of migrant workers' children. The education level of the labor force was a barrier to the adoption of a wage-employed strategy, and wage-employed households shouldered a heavy burden caused by children's education at the same time. For non-labor households, they had a low level of asset possession, and the transfer income-oriented non-labor households were the main object of poverty reduction.

**Author Contributions:** Conceptualization, R.S. and J.M.; methodology, R.S.; software, R.S.; validation, R.S., S.C. and X.G.; formal analysis, R.S.; investigation, R.S.; resources, R.S.; data curation, R.S., S.C. and X.G.; writing—original draft preparation, R.S.; writing—review and editing, R.S. and S.C.; visualization, S.C. and X.G.; supervision, J.M.; project administration, J.M.; funding acquisition, J.M.

**Funding:** This research was funded by the National Natural Science Foundation of China, grant number 71673068 and the National Social Science Foundation of China, grant number 17ZDA030.

**Acknowledgments:** We acknowledge Sun Yat-sen University for providing the data for the study.

**Conflicts of Interest:** The authors declare no conflict of interest.

## Appendix A

**Table A1.** Pairwise comparison of income using the Dunnett T3 method.

| Livelihood Strategy Comparison | Total Income | Agricultural Income | Wages | Operational Income | Property Income | Remittances | Pensions | Relief Funds |
|---|---|---|---|---|---|---|---|---|
| 1 vs. 2 | −58,423.842 (0.000) | 18,395.162 (0.000) | −23,402.048 (0.000) | −53,061.700 (0.000) | | | | |
| 1 vs. 3 | | 20,520.931 (0.000) | | | −1616.107 (0.018) | −3998.494 (0.000) | −9800.787 (0.000) | −1233.897 (0.000) |
| 1 vs. 4 | −26,745.508 (0.000) | 18,459.961 (0.000) | −44,797.924 (0.000) | | | | −212.592 (0.044) | |
| 2 vs. 3 | 60,392.702 (0.000) | 2125.768 (0.015) | 21,351.278 (0.000) | 53,209.684 (0.000) | −1537.043 (0.028) | −3968.186 (0.000) | −9551.891 (0.000) | −1236.909 (0.000) |
| 2 vs. 4 | 31,678.334 (0.000) | | −21,395.876 (0.001) | 53,004.532 (0.000) | | | | |
| 3 vs. 4 | −28,714.368 (0.000) | −2060.970 (0.000) | −42,747.155 (0.000) | | 1484.048 (0.038) | 4008.168 (0.000) | 9588.195 (0.000) | 1218.496 (0.000) |

Note: Results reported mean differences and *p* values below 5% (in brackets).

**Table A2.** Pairwise comparison of capitals using the Dunnett T3 method and the Chi-square test [1,2].

| Livelihood Strategy Comparison | N1 | N2 | N3 | H1 | H2 | H3 | P1 | P2 | P3 | P4 | F2 | S1 |
|---|---|---|---|---|---|---|---|---|---|---|---|---|
| 1 vs. 2 | 1.223 (0.000) | 4.961 (0.000) | | 6.209 (0.000) | −0.540 (0.000) | −0.474 (0.000) | 109.145 (0.000) | −0.177 (0.000) | 52.236 (0.000) | 26.245 (0.000) | 7.393 (0.007) | |
| 1 vs. 3 | 1.442 (0.000) | 5.371 (0.000) | | −12.212 (0.000) | 0.944 (0.000) | 1.484 (0.000) | 30.497 (0.000) | | 51.606 (0.000) | 12.737 (0.000) | 19.064 (0.000) | 1586.300 (0.000) |
| 1 vs. 4 | 0.965 (0.000) | 4.649 (0.000) | | 3.927 (0.000) | −0.455 (0.000) | −0.346 (0.000) | 51.447 (0.000) | −0.088 (0.000) | 106.297 (0.000) | 81.425 (0.000) | 8.245 (0.004) | |
| 2 vs. 3 | | | | −18.421 (0.000) | 1.483 (0.000) | 1.959 (0.000) | 26.009 (0.000) | 0.194 (0.000) | | | | 3003.124 (0.000) |
| 2 vs. 4 | | | | −2.282 (0.034) | | | 15.455 (0.000) | 0.089 (0.000) | | | | 1550.844 (0.050) |
| 3 vs. 4 | −0.476 (0.014) | −0.722 (0.006) | | 16.139 (0.000) | −1.399 (0.000) | −1.831 (0.000) | 20.999 (0.000) | −0.105 (0.000) | | | 7.429 (0.006) | −1452.280 (0.000) |

[1] For continuous variables (including N1, N2, N3, H1, H2, H3, P2 and S1), the Dunnett T3 method was adopted and the results reported mean differences and *p* values below 5% (in brackets). [2] For categorical variables (including P1, P3, P4 and F2), the Chi-square test was adopted and the results reported $X^2$ values and *p* values significant at the adjusted level ($0.05/6 \approx 0.008$; in brackets).

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
