# Peer review of "Classifying Livelihood Strategies Adopting the Activity Choice Approach in Rural China"

_sustainability, doi:10.3390/su11113019_

Round 1

Reviewer 1 Report

The study is generally conducted well, and I believe that the authors conducted the data analysis in an appropriate way. However, the study is very descriptive, and it is hard to decipher how the results can be linked to sustainability themes. Most of the results are predictable and it is not clear how this study can be embedded in studies on livelihoods in China, or general livelihood studies. Therefore, in order for the paper to be considered for publication, I would suggest the following major and minor revisions.

- What are the descriptive analysis of the households selected for this study? Is it representative? How was the sample size calculated? More background information on descriptive statistics should be provided.

- One of the promising aspects of the SLA is the vulnerability context. The word “vulnerability context” is only mentioned once in the paper. Given the data available, how vulnerable are the households to external and internal shocks and stresses, and how does it influence their livelihood strategies?

- Also very little is mentioned about institutional drivers and barriers. The authors conducted an interview via WeChat with one government official (which is too little for doing unstructured or semi-structured interviews), but the institutional barriers are largely ignored. Also the interview is largely dealt with in the discussion section. Whereas, I think this still belongs to the results section. Did the authors conduct an additional policy analysis?

- Some studies on livelihoods in China are described, but we know very little about the state of knowledge on livelihoods in China. Perhaps this could be further discussed in the results section.

- Layout of figure 3 is not clear, and I would recommend language check throughout the document.

Author Response

Dear Reviewer,

Thank you very much for your helpful suggestions and they are vital for the improvement of our paper.

We revised the manuscript accordingly and concrete revisions are presented in the attached Word file.

Thank you again.

Yours sincerely,

Rui Sun

Harbin Institute of Technology

Reviewer 2 Report

In the paper Authors state that classification of livelihood strategies is important for designing effective and targeted poverty-reducing strategies; for this reason, they propose to classify livelihood strategies adopting the activity choice approach and compare differences among income levels, asset endowments, poverty rates and poverty causes of different household clusters to provide bases for the identification of targeted poverty-reducing strategies. The paper could be of a certain interest for readers of Sustainability; however, at the moment the literature review lacks a good identification of the most authoritative papers in poverty analysis, and specially in multidimensional poverty, which has been recognised as fundamental for designing effective and targeted poverty-reducing strategies.

Some suggestions are:

Alkire, S. and Foster, J. (2011a). ‘Counting and Multidimensional Poverty Measurement’. Journal of Public Economics, 95(7–8): 476–487.

Anand S. and Sen A.K. (1997), Concepts of human development and poverty: a multidimensional perspective. Human Development Papers, United Nations Development Programme (UNDP), New York.

Atkinson A.B. (2003), Multidimensional deprivation: contrasting social welfare and counting approaches, Journal of Economic Inequality, 1, pp. 51-65.

Atkinson A.B. and Bourguignon F. (1982), The comparison of multidimensional distributions of economic status, Review of Economic Studies, 49, pp. 183-201.

Betti G., Cheli B., Lemmi A., Verma V. (2008), The Fuzzy Approach to Multidimensional Poverty: the Case of Italy in the 1990s, in Kakwani N., Silber J. (eds.), Quantitative Approaches to Multidimensional Poverty Measurement, Palgrave Macmillan, pp. 30-48.

Author Response

(The authors gave the same response as above.)

Round 2

Reviewer 1 Report

Please check the document for spelling and grammar once more.

Author Response

Dear Reviewer,

Thank you very much for your valuable suggestion.

As directed, we have attempted to conduct a comprehensive language editing of the document and the revised version of the manuscript has since been uploaded.

Thank you again for your helpful suggestion and best wishes for you.

Yours sincerely,

Rui Sun

Harbin Institute of Technology